# Clinical and Product Features Associated with Outcome of DLBCL Patients to CD19-Targeted CAR T-Cell Therapy

**DOI:** 10.3390/cancers13174279

**Published:** 2021-08-25

**Authors:** Sylvain Lamure, François Van Laethem, Delphine De Verbizier, Claire Lozano, Eve Gehlkopf, Jean-Jacques Tudesq, Chris Serrand, Mehdi Benzaoui, Tarik Kanouni, Adeline Quintard, John De Vos, Emmanuelle Tchernonog, Laura Platon, Xavier Ayrignac, Patrice Ceballos, Anne Sirvent, Mickael François, Hanane Guedon, Philippe Quittet, Cedric Mongellaz, Aurélie Conte, Charles Herbaux, Caroline Bret, Naomi Taylor, Valérie Dardalhon, Guillaume Cartron

**Affiliations:** 1Department of Clinical Hematology, CHU Montpellier, 34295 Montpellier, France; s-lamure@chu-montpellier.fr (S.L.); e-gehlkopf@chu-montpellier.fr (E.G.); jean-jacques.tudesq@chu-montpellier.fr (J.-J.T.); t-kanouni@chu-montpellier.fr (T.K.); e-tchernonog@chu-montpellier.fr (E.T.); p-ceballos@chu-montpellier.fr (P.C.); a-sirvent@chu-montpellier.fr (A.S.); h-guedon@chu-montpellier.fr (H.G.); p-quittet@chu-montpellier.fr (P.Q.); c-herbaux@chu-montpellier.fr (C.H.); 2Institut de Génétique Moléculaire de Montpellier, University Montpellier, CNRS, 34293 Montpellier, France; mehdi.benzaoui@igmm.cnrs.fr (M.B.); cedric.mongellaz@igmm.cnrs.fr (C.M.); naomi.taylor@igmm.cnrs.fr (N.T.); 3Department of Biological Hematology, CHU Montpellier, 34295 Montpellier, France; f-vanlaethem@chu-montpellier.fr (F.V.L.); c-bret@chu-montpellier.fr (C.B.); 4Department of Nuclear Medicine, CHU Montpellier, 34295 Montpellier, France; d-de_verbizier@chu-montpellier.fr; 5Department of Immunology, CHU Montpellier, 34295 Montpellier, France; c-lozano@chu-montpellier.fr; 6Clinical Research and Epidemiology Unit, CHU Montpellier, University Montpellier, 34295 Montpellier, France; c-serrand@chu-montpellier.fr; 7Pediatric Oncology Branch, Center for Cancer Research, National Cancer Institute, National Institutes of Health, Bethesda, MD 20892, USA; 8Pharmacy, CHU Montpellier, 34295 Montpellier, France; a-quintard@chu-montpellier.fr; 9Department of Cell and Tissue Engineering, CHU Montpellier, University Montpellier, 34295 Montpellier, France; j-de_vos@chu-montpellier.fr; 10Critical Care Department, CHU Montpellier, 34295 Montpellier, France; l-platon@chu-montpellier.fr (L.P.); mickael-francois@chu-montpellier.fr (M.F.); 11Department of Neurology, CHU Montpellier, INSERM (INM), University Montpellier, 34295 Montpellier, France; x-ayrignac@chu-montpellier.fr; 12Établissement Français du Sang Occitanie, 34295 Montpellier, France; aurelie.Conte@efs.sante.fr; 13Institute of Human Genetics, University Montpellier, CNRS, 34396 Montpellier, France

**Keywords:** CAR T-cell, axicabtagene ciloleucel, tisagenleucleucel, diffuse large B-cell lymphoma, standard-of-care, immune-monitoring, T-cell exhaustion

## Abstract

**Simple Summary:**

Factors impacting the response to CAR T-cell therapies are not fully understood. In this monocentric prospective study, we describe the outcome of 60 patients with relapsed/refractory diffuse large B-cell lymphoma and transformed follicular lymphoma infused with CD19-directed CAR T-cell products, axicabtagene ciloleucel and tisagenlecleucel. We obtained a 40% complete metabolic response and a 27% partial metabolic response with a median progression-free survival of 3.1 months and a median of overall survival of 12.3 months. We also found that age-adjusted IPI at the time of infusion, product features, in vivo expansion, and CAR T-cell exhaustion phenotype were significatively associated with the efficacy of the CAR T-cell therapy.

**Abstract:**

CD19-directed CAR T-cells have been remarkably successful in treating patients with relapsed/refractory (R/R) diffuse large B-cell lymphoma (DLBCL) and transformed follicular lymphoma (t-FL). In this cohort study, we treated 60 patients with axicabtagene ciloleucel or tisagenlecleucel. Complete and partial metabolic responses (CMR/PMR) were obtained in 40% and 23% of patients, respectively. After 6.9 months of median follow-up, median progression-free survival (mPFS) and overall survival (mOS) were estimated at 3.1 and 12.3 months, respectively. Statistical analyses revealed that CMR, PFS, and OS were all significantly associated with age-adjusted international prognostic index (aaIPI, *p* < 0.05). T-cell subset phenotypes in the apheresis product tended to correlate with PFS. Within the final product, increased percentages of both CD4 and CD8 CAR^+^ effector memory cells (*p* = 0.02 and 0.01) were significantly associated with CMR. Furthermore, higher CMR/PMR rates were observed in patients with a higher maximal in vivo expansion of CAR T-cells (*p* = 0.05) and lower expression of the LAG3 and Tim3 markers of exhaustion phenotype (*p* = 0.01 and *p* = 0.04). Thus, we find that aaIPI at the time of infusion, phenotype of the CAR T product, in vivo CAR T-cell expansion, and low levels of LAG3/Tim3 are associated with the efficacy of CAR T-cell therapy in DLBCL patients.

## 1. Introduction

Diffuse large B-cell lymphoma (DLBCL) is the most frequent B-cell lymphoma in adults. Patients with primary refractory or relapsing (R/R) DLBCL have a poor outcome, with a median of 6.2 months overall survival (mOS) [1]. Recently, a chimeric antigen receptor (CAR) T-cell based-immunotherapy approach targeting CD19 has demonstrated improved OS in R/R DLBCL and transformed follicular lymphoma (t-FL) after at least two lines of treatment, [2,3,4] leading to FDA/EMA approval of tisagenlecleucel (tisa-cel, Kymriah^®^, Novartis Pharma, Basel, Switzerland), axicabtagene ciloleucel (axi-cel, Yescarta^®^, Gilead-Kite, Santa Monica, CA, USA), and lisocabtagene maraleucel (liso-cel, Breyanzi^®^, Bristol-Meyers-Squibb, New York, NY, USA).

A CAR is a recombinant, composite trans-membrane receptor recognizing an antigen on the surface of tumor cells through a domain derived from an antibody fragment (ScFv), independent of the major histocompatibility system. Tisa-cel, axi-cel, and liso-cel all target the CD19 surface antigen through the FMC63 ScFv and activate the engineered T-cell via intracellular signaling domains, combining a CD3-ζ domain with either a CD28 (axi-cel) or 4-1BB (CD137, tisa-cel, liso-cel) co-stimulatory domain. While these different CARs exhibit potent anti-tumor activity, ex vivo and murine experiments have suggested that the different co-stimulatory domains modulate CAR T expansion and exhaustion [5,6,7,8].

The efficacy of the three anti-CD19 CAR T-cell therapies approved for the treatment of R/R DLBCL and t-FL has been demonstrated in several phase II trials showing complete metabolic responses (CMR), ranging from 38% to 54%, and a 2-year OS, ranging from 40% to 58% [2,3,4]. Importantly though, these therapies led to grade 3/4 CRS cytokine release syndrome (CRS) in 2–22% of patients and 10–28% of patients suffered from grade 3/4 immune cell-associated neurotoxicity syndrome (ICANS). These toxicities—associated with abnormally high interleukin-6 (IL-6), interferon-γ, tumor necrosis factor, IL-2, GM-CSF, IL-8, IL-10 cytokine secretion, as well as CD19 expression in mural cells in the brain [9,10,11]—represents one factor currently limiting the use of CAR T-cell therapies to certified centers.

Despite important efforts to manage patient toxicities and optimally select patients [12,13], studies from the standard-of-care in US and Europe show that 50% to 60% of patients fail to respond to approved CAR T-cell therapies [14,15,16]. Different factors have been associated with treatment failure such as the lack of early CAR T-cell expansion following infusion [17,18,19,20,21], high tumor burden, and baseline inflammation [14,15,19,22]. Furthermore, immune composition of the final CAR T-cell product may also affect patient outcome; high percentages of CD8 naive-like cells [19] and CD8 memory cells [23] have been associated with more durable responses.

In this study, we evaluated the outcome of R/R DLBCL patients treated with axi-cel or tisa-cel at the University of Montpellier Hospital as a function of their clinical characteristics, CAR T-cell product, and CAR T-cell phenotype following infusion. Apheresis samples as well as infused products were characterized, and all patients were prospectively followed for toxicity, responsiveness, CAR T-cell expansion, and phenotype.

## 2. Materials and Methods

### 2.1. Study Design and Patients

All adult patients receiving axi-cel or tisa-cel between 1 February 2019 and 1 March 2021 for a R/R DLBCL or t-FL were included in this prospective cohort. Ethical and regulatory approvals were obtained from the Comité de Protection des Personnes Ile-de-France IV (ID RCB: 2019-A03032-55). The study is registered in ClinicalTrials.gov (23 August, 2021, NCT04290000).

### 2.2. Data Sources and Assessment of Variables

Histopathological diagnosis was confirmed by the Lymphopath network [24]. The indication for CAR T-cell therapy was confirmed by the regional referee committee after either progression, stable disease, or partial metabolic response on interim 18-FDG glucose positron emission tomography, coupled with low dose CT (PET) after second-line chemotherapy. The infused CAR T-cell product was chosen at the discretion of the Regional Referee Committee. Leukapheresis was performed as per the manufacturer’s pharmacy guidelines (Kite and Novartis). If required, a bridging therapy was applied during the interim. Patients received a lymphodepleting chemotherapy consisting of fludarabine and cyclophosphamide combination. All patients were hospitalized on the day prior to CAR T-cell infusion and remained hospitalized for at least 10 days. All patients received prophylaxis treatment (cotrimoxazole and valaciclovir) for infection prophylaxis. Products were infused 3–4 days following the lymphodepleting chemotherapy, over the course of 30 min. Clinical, imaging, and biological data were recorded for each patient prior to CAR T-cell infusion. After infusion, patients were evaluated for CRS, ICANS [9], infection, myelotoxicity, and associated treatments, clinical and PET responses at one-month (M1), three-month (M3), six-month (M6), and bi-annual intervals. PET were centrally reviewed at the time of analysis and responses were defined by the 2014 Lugano criteria [25]. The best overall response (BOR) was defined as the best response (complete metabolic response: CMR) or partial metabolic response (PMR)) to CAR T-cell within the 6-months following infusion and before any re-treatment for progression. Progression-free survival (PFS) was defined as the time between CAR T-cell infusion and progression, if any, during follow-up. Overall survival (OS) was defined as the time between CAR T-cell treatment and death, if any, during follow-up. The data presented in this study were collected between 1 February 2019 and 1 May 2021.

### 2.3. Flow Cytometry Analyses

Total blood samples were collected in EDTA on the day of apheresis to evaluate the counts of CD4 and CD8 T-cells, B-cells, and NK-cells. The repartition of T-cell subsets was determined as follows: naive as CD45RA^+^CCR7^+^, early post-thymic naive as CD45RA^+^CD31^+^, effector memory as CD45RA^−^CCR7^−^ (EM), terminal effector memory as CD45RA^+^CCR7^−^ (TEMRA), central memory as CD45RA^−^CCR7^+^ (CM), and regulatory as CD4^+^CD25^hi^CD127^low^ (Treg) cells. Samples were acquired on the NAVIOS flow cytometer (Beckman Coulter, Fullerton, United States of America), and analyses were performed using Kaluza software (v1.3).

The phenotype of the infused CAR T-cell product was analyzed for 16 patients. Briefly, product bags were rinsed with PBS containing 2% FBS. To detect CAR^+^ T-cells, mononuclear cells were first incubated with the biotinylated CD19-CAR detection reagent for 15 min at 4 °C, washed with PBS 2% FBS, incubated with a phycoerythrin cyanin-7-conjugated streptavidin, and then stained with fluorochrome-conjugated mAbs directed against several T-cell markers (CD4, CD45RA, CD45RO). Cell acquisition was performed on a BD LSR II-Fortessa (BD Biosciences), and data were analyzed using Diva (BD Biosciences; version 8.0.1) or FlowJo (Tree Star, version 10) softwares.

CAR T-cell pharmacokinetics (PK) were evaluated from PB samples of 27 consecutive patients at the following post-infusion time points: 2 h, and days 1, 4, 7, 10, 14, 18, 21, 28, and 90. Flow cytometry data were acquired on a FACSCanto II flow cytometer (Becton Dickinson, Frankin Lakes, United States of America) and analyzed using FACSDiva software (Becton Dickinson) [26]. CAR expression was evaluated in CD3-gated T-cells. Exhaustion phenotype of CAR T-cells was evaluated by the level of expression of LAG3, Tim3, and PD-1, monitored with appropriate fluorochrome-coupled antibodies. The limit of detection for CAR T-cells was set at 50 events following background removal. All antibodies are listed in Appendix A.

### 2.4. Statistical Analyses

Categorical variables are presented with their associated numbers and percentages and were compared between groups using either χ2 or Fisher tests. Quantitative variables are presented with their means and standard deviation, or median and interquartile range, and were compared via the Student’s *t*-test for variables with a normal distribution or the Wilcoxon Mann–Whitney test. Response to treatment was analyzed via logistic regression; and time to death and time to progression were analyzed via univariate Cox proportional hazards models. The proportional hazards hypothesis was tested, as well as the log linearity of the quantitative variables. A multivariate regression was then performed on response and PFS for potentially predictive variables (variables with a p-value lower than 0.20 in univariate analyses were tested in this model). The collinearity of the variables was tested, and if they were redundant, the most relevant variable was selected based on clinical considerations. Odds and hazard ratios are presented with their 95% confidence intervals. Statistical significance was designated at the conventional two-tailed α level of 0.05 using statistical analysis systems enterprise Guide 8.2 (SAS Institute, Cary, NC, USA) and GraphPad software version 9 (Graph Pad Prism, La Jolla, CA, USA).

## 3. Results

### 3.1. Patient Characteristics

A total of 60 patients received CAR T-cell treatment during the study period (Table 1). Forty-nine patients were treated with axi-cel (82%) and eleven with tisa-cel (18%). Median age was 64 years (range: 18–79) with 30% of patients older than 70 years (*n* = 18). Patient diagnoses included DLBCL (*n* = 43, 71%), t-FL (*n* = 10, 17%), and high-grade B-cell lymphoma (*n* = 7, 12%), including 3% with double hit lymphoma (*n* = 2). The subgroup of these lymphomas was determined by immunohistochemistry according to Hans algorithm; 48% of tumors had a germinal center origin (GC, *n* = 29), 42% had a non-GC origin (*n* = 25), and 10% remained unclassified (*n* = 6). Amongst the 29 patients with GC-DLCBL, 25 were treated with axi-cel and 4 with tisa-cel amongst the non-GC 21 had axi-cel and 4 tisa-cel; and, for the 10 patients with t-FL, 9 received axi-cel and 1 had tisa-cel. A total of 27% of patients received three or more prior lines of treatment (*n* = 16). The median time between apheresis and CAR T-cell infusion was 40 days (range: 30–174) and 90% of the patients underwent a bridging treatment (*n* = 54), generally based on a combination of rituximab with chemotherapy (77%, *n* = 46) or lenalidomide (10%, *n* = 6). At the time of CAR T-cell infusion, 10% of patients (*n* = 6) were in CMR, 30% had stage I/II disease (*n* = 18), and 60% had stage III/IV disease (*n* = 36). Furthermore, 42% of patients had a high LDH (*n* = 25), 8% had a PS ≥ 3 (*n* = 5), and 3% exhibited central nervous system involvement (*n* = 2). According to aaIPI criteria, risk at the time of infusion was designated as low, intermediate-1, intermediate-2, and high for 27% (*n* = 16), 40% (*n* = 24), 28% (*n* = 17), and 5% (*n* = 3) of patients, respectively (Table 1). One patient did not receive lymphodepleting chemotherapy, due to profound lymphopenia.

### 3.2. Toxicities

Following CAR T-cell infusion, 87% of patients experienced grade 1/2 CRS (*n* = 52) and 5% of patients had a grade 3/4 CRS (*n* = 3) (Table 2). A significant percentage of patients also exhibited ICANS; 27% of patients experienced grade 1/2 (*n* = 16); and 12% had grade three/four symptoms (*n* = 7). In the absence of progressive disease, neither CRS nor ICANS was fatal. Tocilizumab was used to treat persistent grade 1 CRS (>72 h) or de novo ≥ grade 2 CRS; 73% of patients received at least one dose of tocilizumab; and 27% of patients received four infusions. Overall, 47% of patients (*n* = 28) received corticosteroids to treat CRS and/or ICANS. Furthermore, 28% of patients (*n* = 17) were admitted to the ICU; four as a preventive measure due to high risk of CRS/ICANS, six for hemodynamic failure, two for respiratory failure, and five for neurological complications. Median ICU stay was 4 days (2–26 days); two patients required norepinephrine infusion, one required dialysis, one required high-flow oxygen therapy, and one required patient required mechanical ventilation. Overall, 33% of patients (*n* = 20) had a documented infection, with 10 infections occurring after the first month of treatment. Documented infections included nine bacterial infections (pneumonitis, colitis, bacteriemia), five invasive fungal infections (candidemia and aspergillosis), and five CMV reactivations. A total of 59 of 60 patients had grade 3/4 cytopenia after lymphodepleting chemotherapy and CAR T-cell infusion. Myelotoxicity was evaluated as a function of transfusion requirements as all patients already presented with myelotoxicity prior to CAR-T therapy. Red blood cell transfusion independence was reached after a median of 76 days (range: 6–654), platelet transfusion independence after a median of 76 days (range: 7–655), and G-CSF independence after a median of 61 days (range: 4–657). We performed an exploratory analysis for factors associated with toxicity and found no correlation between clinical factors and CRS, ICANS, or infections.

### 3.3. Patient Responses

Response to CAR T-cell therapy was evaluated by PET (Figure 1A) and the overall response rate (ORR) at M1 was 58% (*n* = 35); 35% of patients (*n* = 21) exhibited a CMR while 23% (*n* = 14) had a PMR. Of the non-responding patients, two patients died before M1, related to progressive disease (PD) (*n* = 1) and invasive fungal infection (*n* = 1). The remaining 38% of patients (*n* = 23) had no metabolic response (NMR); 13% (*n* = 8) continued with stable disease (SD) and 25% (*n* = 15) with PD. Interestingly, 2 of the 23 patients, who did not exhibit a metabolic response at M1, responded later with PMR and CMR at M3 and M6, respectively. At M3, a response assessment revealed an ORR of 40% (*n* = 24); 25% (*n* = 15) with a CMR and 15% with a PMR (*n* = 9). By this time point, 10 patients (17%) had died. The best overall response (BOR) rate at any point after CAR T-cell infusion was 63% (*n* = 38); 40% with a CMR (*n* = 24) and 23% with a PMR (*n* = 14; Figure 1A). aaIPI at the time of infusion correlated significantly with CMR as a BOR in univariate analysis (*p* = 0.05) and was not found to influence response in a multivariate analysis. A total of 30 patients were treated at progression after CAR T-cell treatment but only 3 patients achieved a CMR; 1 patient treated with brentuximab-vedotin/bendamustin followed by autologous-SCT and 2 patients treated with a combination of lenalidomide and rituximab.

### 3.4. Patient Survival

Median follow-up of the entire cohort was 6.9 months (0.5–26.1 months). The median PFS (mPFS) was 3.1 months with a 29.3% probability of PFS at 12 months (95% CI:17.0–42.8, Figure 1B). Median duration of response for patients who achieved either a CMR or PMR as their BOR was 22 and 3.3 months, respectively (Figure 1C, *p* = 0.02). mOS of the entire cohort was estimated at 12.3 months (95% CI: 32.9–63.1; Figure 1D) and was not reached for patients who achieved a CMR as BOR (Figure 1E, *p* < 0.01). Of the 60 treated patients, 48% died during follow-up (*n* = 29) related to either lymphoma progression (*n* = 26, 89%), secondary acute myeloid leukemia (*n* = 2, 7%), or invasive fungal infection (*n* = 1, 3%). Gender (*p* < 0.01), aaIPI at the time of infusion (*p* < 0.01) and TMTV (*p* = 0.02) on PET assessment before infusion were all significantly associated with PFS in the univariate model (Figure 1F, Appendix A). Because PET data were assessed on different scanners, PET parameters were not included in the multivariate model. By multivariate analysis, PFS was statistically associated with female gender (HR 3.418; 95% CI 1.323–8.829, *p* = 0.01) and aaIPI at the time of infusion (HR 2.020; 95% CI 1.285–3.176, *p* < 0.01) (Table 3). Furthermore, both female gender (*p* = 0.04) and aaIPI at infusion (*p* < 0.01) were significantly associated with longer OS by univariate analysis.

### 3.5. Comparison of Outcome during the 1st and 2nd Years of CD19-Directed CAR T-Cell Therapies

Following certification of our center for CAR T-cell treatments, the characteristics of treated patients evolved (Appendix A). Specifically, while the same numbers of patients were infused in our center during years 1 and 2 (30/year), 43% of patients had received three or more prior lines of treatment (*n* = 13) in year 1 as compared to only 10% of patients during year 2 (*n* = 3; *p* < 0.01). Furthermore, during the first year, the percentage of patients designated as low-risk by aaIPI when the time of infusion was 10% (*n* = 3 of 30) whereas, in the second year, 43% were low-risk (*n* = 13; *p* < 0.01). These differences in patient characteristics translated to differences in outcome; of the 30 patients treated during the first year, only 8 achieved a CMR (27%), whereas 16 of 30 patients treated during the second year achieved a CMR (53%). This was associated with a significant difference in mPFS estimations—2.2 and 6.3 months in years 1 and 2, respectively (*p* = 0.03; Appendix A).

### 3.6. Apheresis Product Characteristics

We first compared the lymphocyte subsets in PB at the time of apheresis with samples from the apheresis bag, finding a comparable composition in the first 18 patients (Appendix A). Further analyses were then performed to address potential correlations between the percentages of different T-cell subsets in PB at the time of apheresis and clinical outcome. Multivariate analysis showed that a high percentage of effector memory (EM) of CD8 T-cells was trended to associate with a better PFS (HR 0.97; 95% CI 0.950–1.001, *p* = 0.06; Table 3). Importantly, evaluation of the different T-cell subsets in the final CAR T-cell product (for both axi-cel and tisa-cel) revealed a significant impact of the manufacturing process on the percentage of T-cell subsets; production was associated with an increase in CD4 T-cells (*p* < 0.01), naive CD8 T-cells (*p* < 0.01), CM CD4 and CM CD8 cells (*p* < 0.01 each). Conversely, there was a decrease in EM CD4 (*p* < 0.01), EM CD8 (*p* = 0.05) and EMRA CD4 cells (*p* = 0.02, Figure 2).

### 3.7. Phenotype of the CAR T-Cell Final Product

We immunophenotyped the few cells remaining in the CAR T-cell bags following their infusion into patients (15 axi-cel, 1 tisa-cel). Importantly, there was a high heterogeneity in the percent transduction, with CD19 CAR expression detected in 22.6% to 78.1% of cells (median: 57.8%; IQR 49.8–66.5%). We then compared the immune subset distribution between the CAR^−^ and CAR^+^ T-cells. Comparison of the phenotypes between CAR^+^ and total T-cells revealed similar CD4/CD8 ratios and similar percentages of the different subsets (naive-like/CM/EM/EMRA, Appendix A). We further analyzed the correlation between the different CAR^+^ T-cell subsets and the clinical response (Figure 3) and found that enrichment of the final product in EM CD4/CD8 CAR^+^ T-cells together with a lower percentage of naive-like CD4/CD8 CAR^+^ T-cells was associated with CMR.

### 3.8. In Vivo Pharmacokinetics of CAR T-Cells

Following infusion, CAR^+^ T-cells were detected in the PB of 27 patients: 23 received axi-cel and 4 received tisa-cel. Because CAR T-cell monitoring could not be performed on all treated patients, CAR T-cell PK were not included in multivariate analyses. The maximal concentration (Cmax) of CAR^+^ CD3 T-cells was detected on day 7 (Tmax range: 7–14) and ranged from 15% to 92% of total lymphocytes in 25 of 27 patients (Figure 4A). The two patients who showed a Cmax value of <10% exhibited disease progression (*p* = 0.06). Importantly, and in line with previous studies from the literature [17,18,19,20,21], high numbers of CAR^+^ T-cells were detected in all patients exhibiting CMR/PMR. More precisely, a higher Cmax and area under the curve (AUC_0-28_) of CAR^+^ T-cells were both significantly associated with CMR/PMR; the median Cmax was 43% (IQR 26–67%) and 27% (IQR 10–36%; *p* = 0.04), while the median AUC_0-28_ was 370 (IQR 225–652) and 92 (IQR 24–213) in responders and non-responders (*p* = 0.03), respectively (Figure 4A). Similar results were found when evaluating the AUC_0-28_ of CD4 CAR^+^ T-cells, with values of 122 (IQR 92–206) in patients with CMR/PMR vs. 47 (IQR 8-929) in NMR (*p* = 0.03, Figure 4B). A higher Cmax of CD8 CAR^+^ T-cells was also significantly associated with CMR/PMR (7% (IQR 4–18%) vs 26% (IQR 13–38%) in responders and non-responders, respectively; *p* = 0.01, Figure 4C). A similar trend was detected when evaluating absolute numbers of CAR^+^ T-cells. However, it is important to note that these latter analyses were challenging because of the lymphopenic status of the treated patients, and the association was not statistically significant (Appendix A).

### 3.9. Evaluation of Exhaustion Phenotype as a Function of Responsiveness to CAR T-Cell Therapy

Exhaustion has been highlighted as an important factor in the responsiveness to CAR T-cell therapies [5,23,27]. Therefore, we evaluated the expression of the PD-1, LAG3, and Tim3 exhaustion markers on CAR^+^ T-cells collected at D7, D10, D14, and D18 post-infusion (Figure 5A). The most significant differences were observed for LAG3 expression at day 10, and a lower percentage of LAG3^+^ CD4 CAR^+^ T-cells was associated with CMR/PMR (*p* = 0.01) Similarly, at day 10, the percentages of LAG3^+^ CD8 CAR^+^ T-cells and Tim3^+^ CD8 CAR^+^ T-cells were lower in patients with CMR/PMR as compared to non-responders (*p* = 0.04; Figure 5B). Interestingly, the LAG3^+^ percentage was similar on CAR^−^ and CAR^+^ T-cells, whereas the Tim3^+^ percentage was lower on CAR^−^ than CAR^+^ T-cells (Appendix A). Finally, the frequency of PD-1^+^ cells within the CAR^−^ and CAR^+^ subsets did not change in any condition or time point; as such, it did not represent a discriminating parameter in our study (Figure 4C and Appendix A). Thus, our study shows that the upregulation of LAG3 and Tim3 on CAR^+^ T-cells by day 10 following CAR T-cell infusion is negatively correlated with CMR/PMR.

## 4. Discussion

In this study, we report the outcome of 60 patients with R/R DLBCL and t-FL who were consecutively treated with commercial CAR T-cells in a single university center. We prospectively collected and evaluated the immune composition of apheresis samples and CAR T-cell products, and monitored the in vivo PK of CAR T-cells after infusion. The BOR rate was 63%, including 40% CMR, with a mOS of 12.3 months, and we found that aaIPI at infusion correlated with CMR, PFS, and OS. Moreover, we described that the in vivo level of CAR T-cell exposure and a low level of markers of exhaustion phenotype was significantly associated with the metabolic response rate.

The rates of CMR at M1 (37%) and M3 (25%) in this study appear lower than those detected in pivotal phase II trials [3,28] and standard-of-care studies [16,29]. However, comparisons between studies are challenging because of variations in the definition of BOR, time for assessment, and patient characteristics prior to treatment. After a short FU of 6.9 months, we observed a mOS and mPFS of 12.3 and 3.1 months, respectively—similar to those reported in the Juliet trial (mOS: 12 months, mPFS: 3 months) [3] but lower than the values reported in the ZUMA-1 trial (mOS > 24 months, mPFS: 5.9 months) [28]. Our survival results are similar to those reported in a retrospective monocentric study [14] but lower than trials with longer FU [15] and higher numbers of patients [16,29]. This may be due to patient eligibility criteria; a study comparing ZUMA-1-eligible and non-eligible patients showed higher CMR, PFS, and OS in the former [30]. Indeed, PS ≥ 2 and the use of bridging therapy were exclusion criteria in ZUMA-1. As indicated above, aaIPI at the time of infusion correlated with CMR, PFS, and OS in our study but, notably, almost all of our patients received a bridging therapy. This was due, at least in part, to a longer median time for CAR T-cell production; 40 days compared to 17 days in the ZUMA-1 trial. Furthermore, 8% of our patients had a PS 3-4 at the time of infusion, a criterion that would have made them ineligible for the ZUMA-1 trial. These differences may, at least partially, explain the lower rates of response obtained in our study, wherein the majority of patients (82%) received axi-cel, as compared to the ZUMA-1 trial. That being said, our CMR and PFS rates increased between the first and second years of activity, likely because patients treated during the first year presented with greater risk factors—a higher percentage of patients treated in year one had received >three lines of treatment and had a higher aaIPI at the time of infusion. These data highlight the critical role of patient selection criteria and, potentially, the manufacturing time of the CAR T-cell product for the success of CAR T-cell therapy [13].

The immune composition of the CAR T-cell product is heterogeneous and recent data strongly suggest that this heterogeneity influences T-cell fitness and clinical outcome [19,20,23,31]. This heterogeneity is impacted by both patient-specific characteristics as well as the manufacturing process [19,32,33,34]. Our results suggest that the enrichment of the product in CD4/CD8 CAR^+^ cells with an EM phenotype and a lower percentage of naive-like cells may be associated with an improved response to treatment. Notably, using a single-cell RNA sequencing approach, comparing axi-cel CAR T-cell products from patients who went into a complete remission with those who did not, Deng et al. correlated CMR with a three-fold higher frequency of CD8-expressing memory signatures. Locke et al. described distinct results on the ZUMA-1 cohort; flow cytometry analyses revealed that high percentages of naive-like (CCR7+CD45RA+) T-cells and a lower ex vivo doubling time was associated with a higher response rate [19,35,36]. In future studies, it will be important to determine whether the ex vivo culture conditions—including T-cell selection, stimulation methods, media composition, and cytokine cocktails—may impact the final product composition and its clinical outcome [31,33].

CAR T-cell toxicities are strongly related to the vector composition as higher rates of grade ≥ 3 CRS, and ICANS were observed in patients treated with axi-cel, compared to tisa-cel and liso-cel [2,3,4]. In our study, in which 82% of the patients were treated with axi-cel, we observed a rate of 5% and 12% of grade ≥ 3 CRS and ICANS, respectively. These percentages are lower than those reported in the ZUMA-1 trial [2] and standard-of-care studies using axi-cel [16,30], likely related to the improved management of patients. Our early administration of tocilizumab and corticosteroids and the preemptive transfer of a significant number of patients to the ICU resulted in a lower level of non-manageable toxicities. As regards infections as a CAR T-cell-related toxicity, five patients in our cohort developed invasive fungal infections (candidemia and aspergillosis) and five exhibited CMV reactivations. This level could be associated with tocilizumab and corticosteroids treatments [37].

In our cohort, CMR was associated with a higher AUC_0-28_ and Cmax of CAR^+^ CD3 T-cells. This is consistent with reports from different studies, describing that a peak in the peripheral concentration of CAR T-cells at 7 to 10 days after infusion correlates with an objective response rate [2,17,18,20,21,28]. The long persistence of CAR T-cells in patients has also been demonstrated in the ZUMA-1 trial, with 66% of patients harboring detectable CAR-expressing T-cells at 2 years post-infusion [38]. The time between infusion time and Cmax, as well as the long persistence of CAR T-cells in PB, indicate an in-vivo expansion and long-term survival of CAR T-cells in patients, as demonstrated for antigen specific T-cells after TCR stimulation [39]. The relationship between the early expansion of CAR T-cells and the clinical response may be influenced by tumor burden, as recently suggested [19]. As such, TMTV [15] and disease metabolic volume kinetics can serve as a prognostic tool before CAR T-cell infusion [14]. These results underline the importance of maintaining a favorable target-to-effector ratio, at least during the first month of treatment, and point to the importance of patient selection criteria and the CAR T-cells dosing regimen.

From a technical perspective, different methods have been used to monitor PB CAR T-cells in patients, based either on PCR [40,41,42] or flow-cytometry [43,44]. As compared to PCR-based approaches, flow-cytometry CD4 and CD8 CAR^+^ subpopulations can be distinguished [44]. Expression of the exhaustion phenotype has been described to correlate with early molecular failure after CAR T-cell infusion [5,23,27]. In our study, high-throughput flow cytometry analyses were used to evaluate the overall dynamics of exhaustion using surface markers on these subsets—TIM3, LAG3, and PD1. PD-1 expression was not correlated with a clinical response. Notably though, at D10 following infusion, a high percentage of CD4 and CD8 CAR^+^ T-cells expressing LAG3 as well as a high percentage of CD8 CAR^+^ T-cells expressing Tim3 were associated with the lack of a metabolic response. These data are of much importance as they present a very early readout that is associated with responsiveness to CAR T-cell therapy.

## 5. Conclusions

Our study reports on the outcome of 60 R/R DLBCL and t-FL patients treated with axi-cel or tisa-cel at the University Hospital Center of Montpellier. Prospective evaluation demonstrated the influence of aaIPI on the clinical response and outcome as well as the importance of patient selection and in vivo CAR T-cell expansion. Furthermore, our analyses of product characteristics, conditioned by the apheresis content, strongly suggest that this heterogeneity impacts patient outcome. Thus, protocols that optimize the phenotype of the CAR T-cell product, together with improved patient selection, will be critical in optimizing the efficacy of CAR T-cell therapy in DLBCL patients.

## Figures and Tables

**Figure 1 cancers-13-04279-f001:**
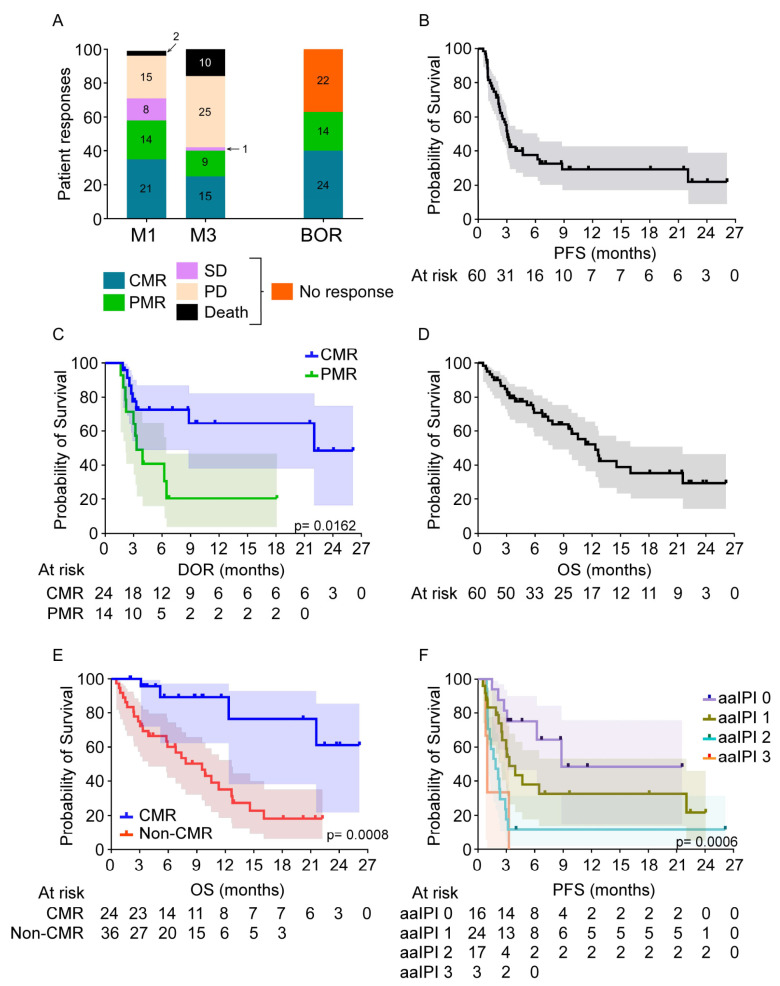
Clinical responses and survival of DLBCL patients treated with CD19-targeted CAR T-cell therapy. (**A**) Responses of patients at 1 (M1) and 3 months (M3) following CAR T-cell infusion as well as best overall response (BOR). The percentages and absolute numbers of patients with a complete (CMR) or partial (PMR) clinical response, stable disease (SD), and progressive disease (PD) are presented. (**B**) Progression-free survival (PFS) of the entire cohort (n:60) is presented as a function of time. (**C**) Duration of response (DOR) is presented as a function of time. (**D**) Overall survival (OS) of the entire cohort (n:60) is presented as a function of time. (**E**) OS following CART-cell therapy is presented as a function of the presence or absence of a CMR. (**F**) PFS following CAR T-cell therapy is presented as a function of the age-adjusted international prognostic index (aaIPI) at the time of infusion. Response to treatment was analyzed via logistic regression; time to death and time to progression were analyzed via univariate Cox proportional hazards models; and p-values were calculated using Wilcoxon Mann Whitney test.

**Figure 2 cancers-13-04279-f002:**
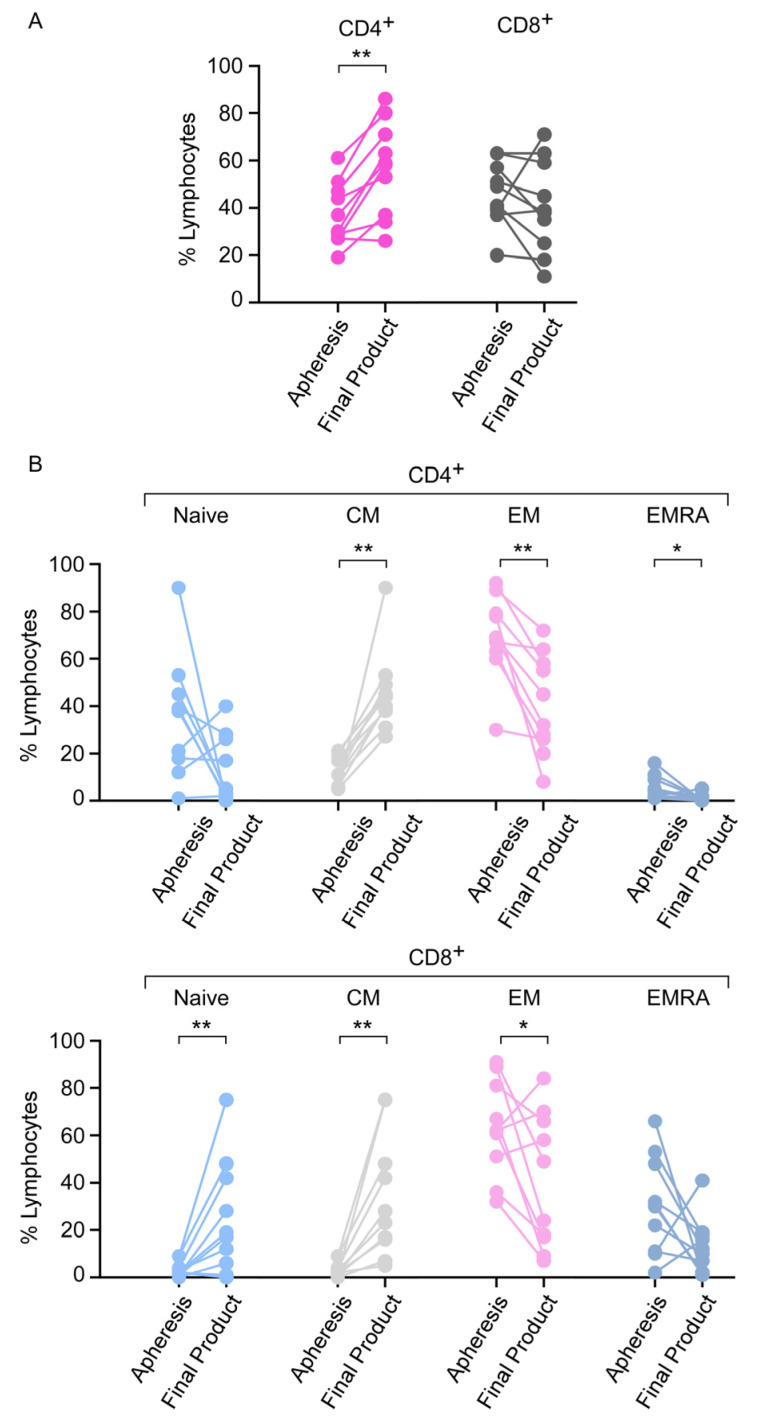
Percentages of the different T-cell subsets in the apheresis and commercial CAR T-cell products. (**A**) Percentages of CD4 and CD8 T lymphocytes in the apheresis and final products are presented for 10 patients. (**B**) The phenotypes (naive, central memory (CM), effector memory (EM), effector memory re-expressing CD45RA (EMRA)) of CD4 and CD8 T-cells in the apheresis and CAR T-cell products are presented. *p*-values were calculated using a Mann–Whitney test and significant values are shown (*, *p*-value between 0.02 and 0.05; **, *p* < 0.02).

**Figure 3 cancers-13-04279-f003:**
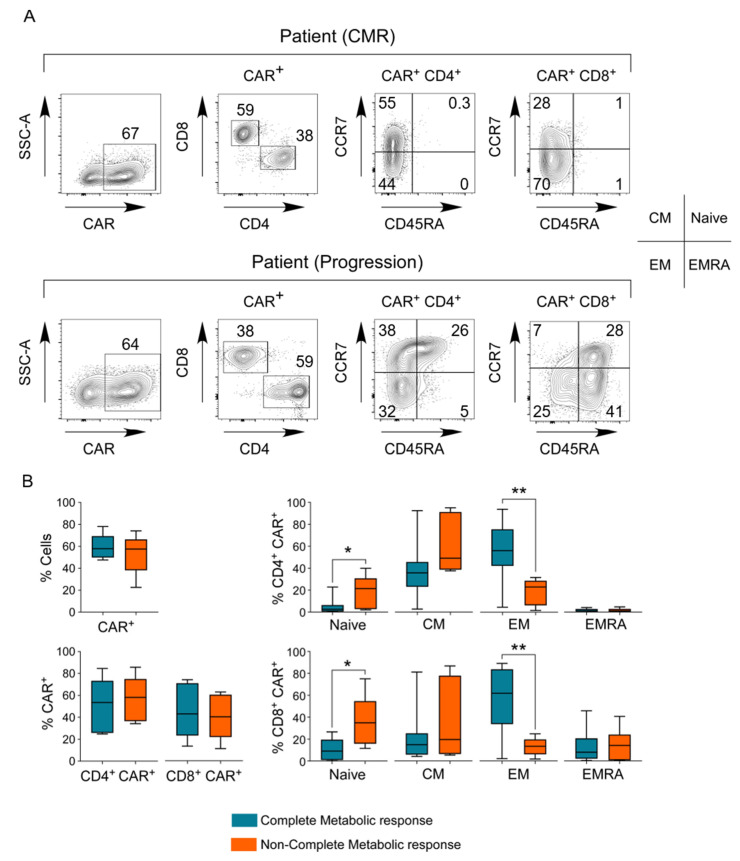
Phenotypes of the CAR T-cell products and correlation with the clinical response. (**A**) Representative dot plots of the CAR T-cell products from one patient with a CMR (above) and one patient with progressive disease (below) are shown. (**B**) Quantification of the percentages of CAR^+^ T-cells in patients who achieved a CMR and patients who did not (top left). Percentages of CD4^+^CAR^+^ and CD8^+^CAR^+^ T-cells in the two groups are presented (bottom left). The percentages of naive, central memory (CM), effector memory (EM), effector memory re-expressing CD45RA (EMRA), and CD4^+^CAR^+^ and CD8^+^CAR^+^ T-cell subsets were determined, and the median, IQR, and range (n:16) are presented. *p*-values were calculated using a Mann Whitney test and significant values are shown (*, *p*-value between 0.02 and 0.05; **, *p* < 0.02).

**Figure 4 cancers-13-04279-f004:**
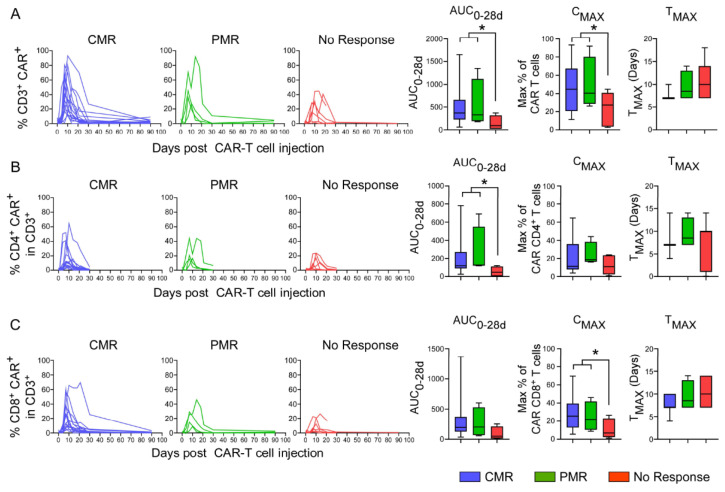
In vivo pharmacokinetics of CAR^+^ T-cells following infusion. (**A**) The percentages of CAR^+^ T-cells within the peripheral CD3^+^ lymphocyte pool were monitored in 27 patients by flow cytometry at the indicated time points following infusion and are presented as a function of their clinical response; CMR, PMR, and no response (left). Area under the curve (AUC_0-28_), representing the area under the curve of the percentages of CAR^+^ T-cells from day 0 to day 28 post-CAR^+^ T infusion; maximal concentration post-infusion (C_MAX_); and time to maximal concentration (T_MAX_), and their median, IQR, and range (n:27), are presented. (**B**) The percentages of CD4^+^ CAR^+^ cells within the total peripheral CD3^+^ T-cell population are presented as in panel A, as well as AUC_0-28_, C_MAX_, and T_MAX_ are presented. (**C**) Evaluations are presented as a function of CAR^+^ CD8^+^ T-cells. Note that patients with incomplete kinetics were not included in the AUC evaluation. *p*-values were calculated using a Mann–Whitney test and significant values are shown (*: *p* < 0.05).

**Figure 5 cancers-13-04279-f005:**
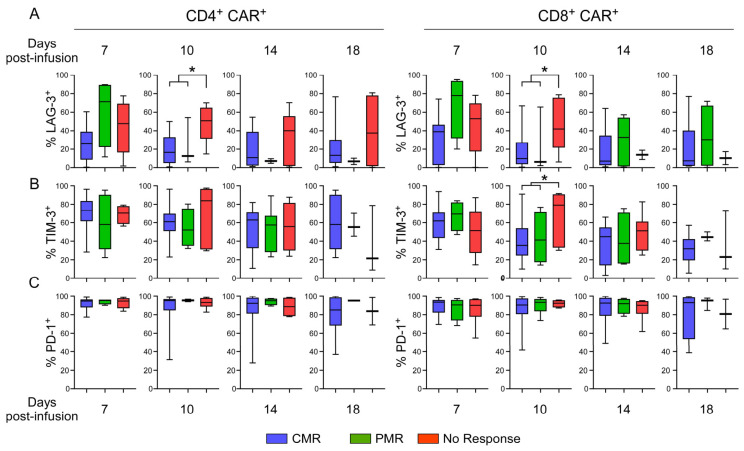
Evaluation of the exhaustion phenotype of CAR^+^ T-cells following infusion. Exhaustion of CD4^+^ CAR^+^ (left) and CD8^+^ CAR^+^ (right) T-cells was evaluated from days 7 to 18 following CAR T infusion as a function of LAG3 (**A**) TIM-3 (**B**) and PD-1 (**C**) expression. Expression was assessed in patients with a CMR (n:13), PMR (n:4), and no response (n:6), and their median, IQR, and range (n:27) are presented. *p*-values were calculated using a Mann–Whitney test and significant values are shown (*: *p* < 0.05).

**Table 1 cancers-13-04279-t001:** Baseline characteristics of patients.

Characteristics	All Population
	(*n* = 60)
Demography	
Age, years	
Median (range)	64 (18–79)
≥70, *n* (%)	18 (30)
Male gender, *n* (%)	38 (63)
Lymphoma, n (%)	
Histology	
DLBCL	43 (71)
Transformed follicular lymphoma	10 (17)
High grade DLBCL	7 (12)
Number of previous line therapy	
≤2 lines	44 (73)
>3 lines or more	16 (27)
Autologous SCT	12 (20)
Bridging therapy, *n* (%)	54 (90)
Characteristics at infusion	
Ann Arbor stage, *n* (%)	
No measurable disease	6 (10)
I/II	18 (30)
III/IV	36 (60)
LDH > normal limit, *n* (%)	25 (42)
PS 3–4, *n* (%)	5 (8)
Median time between apheresis and infusion, days (range)	40 (30–174)
aaIPI, *n* (%)	
Low	16 (27)
Intermediate-1	24 (40)
Intermediate-2	17 (28)
High	3 (5)

Abbreviations: DLBCL Diffuse Large B-Cell Lymphoma, CAR T-cells: Chimeric Antigen Receptor T-cells, SCT Stem Cell Transplantation, LDH Lactate Dehydrogenase, aaIPI age adjusted International Prognosis Index.

**Table 2 cancers-13-04279-t002:** Toxicities after CAR T-cell treatment.

Toxicities and Related Treatments	All Population
	(*n* = 60)
Cytokine-Releasing Syndrome, *n* (%)	
No CRS	5 (8)
CRS I/II	52 (87)
CRS III/IV	3 (5)
Immune Cell-associated Neurotoxicity, *n* (%)	
No ICANS	37 (62)
ICANS I/II	16 (27)
ICANS III/IV	7 (11)
Treatment of CRS & ICANS, *n* (%)	
Tocilizumab	44 (73)
Corticosteroids	28 (47)
Intensive care unit transfer, *n* (%)	17 (28)
Hemodynamic failure requiring amines	2 (3)
Renal failure requiring extracorporeal epuration	1 (2)
Respiratory failure requiring high-flow O_2_	1 (2)
Neurologic failure requiring ventilation	1 (2)
Documented infections, *n* (%)	20 (33)
After M1	10 (17)
Documented bacterial infection	9 (15)
Invasive fungal infection	5 (8)
CMV reactivation	5 (8)
Myelotoxicity	
RBC transfusion independency, days (median, range)	76 (6–654)
>3-months RBC transfusion, *n* (%)	27 (45)
Platelet transfusion independency, days (median, range)	76 (7–655)
>3-months platelets transfusion, *n* (%)	24 (40)
G-CSF withdrawal, days (median, range)	61 (4–657)
>3-months G-CSF treatment, *n* (%)	20 (33)

Abbreviations: CRS—cytokine-releasing syndrome, ICANS—immune cell-associated neurotoxicity, RBC—red blood cells, G-CSF—granulocyte colony-stimulating factor.

**Table 3 cancers-13-04279-t003:** Multivariate analysis of factors associated with PFS after CAR T-cells therapy.

Factors	Adjusted *p*-Value	Hazard Ratio (IC95%)
Male Gender vs. Female (reference)	0.0112	3.418 (1.323–8.829)
aaIPI	0.0023	2.020 (1.285–3.176)
%CD8 effector memory in apheresis product	0.0643	0.975 (0.950–1.001)

Due to missing data, PET characteristics as well as product attributes and CAR T-cells PK were not included in multivariate analysis. Abbreviations: DLBCL—diffuse large B-cell lymphoma, aaIPI—age-adjusted international prognosis index, PET—positron emission tomography, PK—pharmacokinetics.

## Data Availability

The dataset for this study can be provided on demand by contacting the corresponding authors.

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
