# Peer review of "Clinical and Product Features Associated with Outcome of DLBCL Patients to CD19-Targeted CAR T-Cell Therapy"

_cancers, 2021, doi:10.3390/cancers13174279_

Round 1
Reviewer 1 Report
The manuscript "Clinical and product feature associated with outcome of DLBCL patients to CD19-targeted CAR- T-cell therapy" by Sylvain Lamure and colleagues evaluate the efficacy of axicabtagene ciloleucel and tisagen-lecleucel in treating 60 patients with relapsed/refractory diffuse large B-cell lymphoma (DLBCL) and transformed follicular lymphoma (t-FL). The researchers obtained a complete and partial response in 40% and 23% of patients, respectively, and a median progression-free survival (mPFS) of 3.1 months and overall survival (mOS) of 12.3 months, respectively, confirming the already known promising role of this targeted therapy in these lymphomas.
This study confirms the promising role of the CD19-targeted CAR T-cell therapy, already showed from some phase 2 trials, and find some correlation with prognosis, phenotype of the CAR T product end level of the LAG3 and Tim3.
The study deals with a very current and exciting topic, and although well structured methodologically, it enrols a limited number of patients, of whom an overall short follow-up is available.
Comments
The authors state that they enrol 60 patients, of which 43 diffuse large B cell lymphomas (DLBCL), ten transformed follicular lymphomas (t-FL) and seven high-grade lymphomas (HG-L); please, clarify the sub-group of these tumours (t-FL, GC- or activated-DLBCL), specifying the presence of MYC, BCL2 and BCL6 rearrangements.
The authors state that 49 patients received axi-cel and 11 with tisa-cel; please specify how many t-FL, GC- or activated DLBCL have been treated with axi-cel or tisa-cel, respectively and show if there were different results in the respective responses to treatment.
The text is often non-flowing and challenging to read, especially in the results section; it would benefit from an extensive review for clarity.
Author Response
Dear editor, Dear reviewers,
We would like to thank you for your careful and helpful review of our manuscript.
In the revised version of the manuscript, we added the sections “simple summary”, “conclusion” and “back matters” as requested by the editor. We also revised the manuscript according to the reviewers’ comments.
You will find below a detailed response to the important comments raised by the reviewers and revisions in the text are highlighted to facilitate the review.
Kind regards,
Sylvain Lamure, Valérie Dardalhon and Guillaume Carton
Reviewer 1
The authors state that they enrolled 60 patients, of which 43 diffuse large B-cell lymphomas (DLBCL), ten transformed follicular lymphomas (t-FL) and seven high-grade lymphomas (HG-L); please, clarify the sub-group of these tumors (t-FL, GC- or activated-DLBCL), specifying the presence of MYC, BCL2 and BCL6 rearrangements.
We thank the reviewer for her/his comment and pertinent review of our manuscript. We apologize for the lack of precision and we have now added information regarding the patients’ subgroups in section 3.1 “Patient characteristics.” The Germinal Center (GC)/non-GC status was determined using Hans algorithm, which does not always reproduce faithfully the genotype analysis: In our study, 29 patients had a GC phenotype, 25 had a non-GC phenotype and 6 remained unclassified. Furthermore, two patients had high-grade Double Hit Lymphoma (DHL) Lymphoma according to WHO classification (section 3.1 line 190-192).
The authors state that 49 patients received axi-cel and 11 with tisa-cel; please specify how many t-FL, GC- or activated DLBCL have been treated with axi-cel or tisa-cel, respectively and show if there were different results in the respective responses to treatment.
Amongst the 29 patients with GC-DLCBL, 25 were treated with axi-cel and 4 with tisa-cel, amongst the non-GC 21 had axi-cel and 4 tisa-cel and for the 10 patients with t-FL, 9 received axi-cel and 1 had tisa-cel. The low number of patients treated with tisa-cel does not allow for statistical analysis and a robust statistical comparison between the 2 treatments. However, we performed multivariate analysis according to histologic subgroups, including the GC/nonCG classification (suppl table 2) and no statistical differences were identified between axi-cel and tisa-cel.
The text is often non-flowing and challenging to read, especially in the results section; it would benefit from an extensive review for clarity.
We apologize for this lack of clarity and we have revised the results section to improve the description of the data presented in our study.
Reviewer 2 Report
This is a clearly written report on their experience of treating 60 patients R/R DLBCL tFL with CART axi cell and tisa cell in one French center with commercially available products. With a short follow-up, CMR were obtained in 40% of cases with 23%PMR and mOS was 12 months. These results are slightly inferior to those reported before in the US. They looked at factors affecting survival. Of interest, they were able to follow CART cell products and CART cell phenotype. Increase percentage of CD4/CD8 in apheresis product were associated with a better PFS. CMR was associated with higher in vivo expansion of CAR T and lower expression of LAG3 and Tim3, markers of exhaustion. saaIPI was associated with CMR, PFS and OS.
Mechanisms associated with durable response remain incompletely elucidated. Prespecified clinical covariates were not clearly predictive of clinical efficacy. The analysis of biomarkers is on going with different results. It is important to increase the number of studied cases to better evaluate the results obtained with CAR T salvage approach.
Comments:
Looking at markers in a cohort of patient need to include most of the patients. Unfortunately, only 16 pts were evaluable for phenotype and 27 for PK analysis. Consequently, it is difficult run a global analysis on the 60 patients with their clinical covariates. (Table 3)
Page 3, criteria for inclusion, protocol description should be better summarized.
aaIPI was estimated at the time ( before?) of infusion. Is it similar to the secondary aa IPI at relapse or progression? It can be confusing as your patients had bridging therapy described in several studies as a negative factor. This factor is not always found in other series or replaced by MTV which could be estimated retrospectively?
The population remains quite heterogenous with 30% over 70 years. Were younger patient’s ineligible to autologous SCT, only 12 performed.?
High grade lymphoma; how many had DHIT?
Considering the rate of infections or SAE, did you find a relation with age? Did you have prophylaxis for infections.
Myelotoxicity is presented according to their supportive care. Can you provide the classical time to recovery?
The second part of the article with figures on phenotypes and PK of CART are interesting but the limited number of patients studied and the short follow up, does not allow robust statistic and could be reduced to the main findings.
Author Response
Dear editor, Dear reviewers,
We would like to thank you for your careful and helpful review of our manuscript.
In the revised version of the manuscript, we added the sections “simple summary”, “conclusion” and “back matters” as requested by the editor. We also revised the manuscript according to the reviewers’ comments.
You will find below a detailed response to the important comments raised by the reviewers and revisions in the text are highlighted to facilitate the review.
Kind regards,
Sylvain Lamure, Valérie Dardalhon and Guillaume Carton
Reviewer 2
Looking at markers in a cohort of patient need to include most of the patients. Unfortunately, only 16 pts were evaluable for phenotype and 27 for PK analysis. Consequently, it is difficult run a global analysis on the 60 patients with their clinical covariates. (Table 3)
We appreciate the reviewer’s critique and agree that our PK analysis and phenotyping of CAR T-cell final products in 27 and 16 patients, respectively, are exploratory. However, sampling was performed on patients recruited consecutively, thus limiting the risk of selection bias. Furthermore, even with this number of patients, we detected a significant correlation between some of the parameters tested (T cell phenotype, exhaustion level, etc..) and the clinical response associated with CAR T-cell treatment (Figure 3 and supplementary Figure 3). We also provide important information on the immune composition at different steps of CAR T cell therapy––from apheresis to in vivo follow up after CAR T cell infusion. We therefore found it important to present these data to the community and appreciate your input.
The data presented in Table 3 represent multivariate analysis on PFS. As a result of the low number of samples analysed, we could not test the influence of PK and phenotype of the CAR T-cell product (available for 27 and 16 patients as correctly pointed out by the reviewer) on PFS. However, we did find that the percentage of CD8 effector memory cells in the apheresis product tended to influence PFS (analysis performed on all patients, n=60; Table 3).
Page 3, criteria for inclusion, protocol description should be better summarized.
We apologize for the lack of clarity regarding the protocol description and have now revised this section (pages 4 &5).
aaIPI was estimated at the time (before?) of infusion. Is it similar to the secondary aa IPI at relapse or progression? It can be confusing as your patients had bridging therapy described in several studies as a negative factor. This factor is not always found in other series or replaced by MTV which could be estimated retrospectively?
We thank the reviewer for this question: in our study, aaIPI was determined at the time of CAR-T cell infusion (this aspect is now clarified in the revised manuscript). TMTV is an important prognostic factor in DLBCL and we find that TMTV at infusion correlated with PFS in univariate analysis (Section 3.4 “Patient survival”). However, we did not include TMTV in our multivariate analysis because PET data was acquired on different scanners (Table 3).
The population remains quite heterogenous with 30% over 70 years. Were younger patient’s ineligible to autologous SCT, only 12 performed?
In the presented cohort, 33 of 43 patients who were younger than 70 years of age did not reach CMR after second line of treatment with 2 cycles of R-DHAC. They were therefore included in the CAR T-cell program. Moreover, younger patients were eligible for ASCT as the cut-off in our center is 65 years of age. In our cohort, 55% of treated patients (n=33) were younger than 65 years and 23 did not reach CMR after second line of treatment with R-DHAC.
High grade lymphoma; how many had DHIT?
Two patients had DHL high-grade lymphoma according to WHO classification. This information is now added to the manuscript, section 3.1 of the results.
Considering the rate of infections or SAE, did you find a relation with age? Did you have prophylaxis for infections.
All patients received prophylaxis treatment with valaciclovir and cotrimoxazole until CD4 T-cell repopulation and at least for one year after CAR T-cell infusion. This information is now added to the text; “Data sources and assessments variable” section (line 109). We performed an exploratory analysis for factors associated with toxicity and found no correlation between age and CRS, ICANS or infections. This information is now added in the section 3.2 “toxicities”.
Myelotoxicity is presented according to their supportive care. Can you provide the classical time to recovery?
We appreciate this critique and struggled with this point. In our cohort, all patients presented with myelotoxicity as a result of previous lines of treatment and for many patients, complete blood counts at follow up were performed outside the hospital, introducing a potential source of bias. As such, we used recorded transfusion and g-csf requirements as a uniform parameter for evaluating myelotoxicity following CAR T cell therapy. This is now clarified on page 6, lines 425-426.
The second part of the article with figures on phenotypes and PK of CART are interesting but the limited number of patients studied and the short follow up, does not allow robust statistic and could be reduced to the main findings.
We appreciate and agree with the reviewer’s comment that our biological findings on 16 and 27 patients are exploratory. However, as detailed in our response above, we thought it important to present these data to the community, especially as some of the detected alterations significantly correlated with outcome. We have though consolidated these data and limited their presentation.
Reviewer 3 Report
Congratulations on this work. No further comments
Author Response
Dear editor, Dear reviewers,
We would like to thank you for your careful and helpful review of our manuscript.
In the revised version of the manuscript, we added the sections “simple summary”, “conclusion” and “back matters” as requested by the editor. We also revised the manuscript according to the reviewers’ comments.
You will find below a detailed response to the important comments raised by the reviewers and revisions in the text are highlighted to facilitate the review.
Kind regards,
Sylvain Lamure, Valérie Dardalhon and Guillaume Cartron
Rewiever 3
Congratulations on this work. No further comments
We appreciate the reviewer’s congratulations and we thank her/him for this kind comment.